# Substances of Interest That Support Glaucoma Therapy

**DOI:** 10.3390/nu11020239

**Published:** 2019-01-22

**Authors:** Sergio Claudio Saccà, Paolo Corazza, Stefano Gandolfi, Daniele Ferrari, Samir Sukkar, Eugenio Luigi Iorio, Carlo Enrico Traverso

**Affiliations:** 1Ophthalmology Unit, Department of Head/Neck Pathologies, Policlinico San Martino Hospital, IRCCS Hospital-University San Martino, Viale Benedetto XV, 16132 Genoa, Italy; ferraridaniele92@gmail.com; 2Eye Clinic, Department of Neuroscience and Sensory Organs, University of Genoa, Policlinico San Martino Hospital IRCCS Hospital-University San Martino, Viale Benedetto XV, 16132 Genoa, Italy; polcorazza@gmail.com (P.C.); mc8620@mclink.it (C.E.T.); 3Ophthalmology Unit, Department of Biological, Biotechnological and Translational Sciences, University of Parma, 43121 Parma, Italy; s.gandolfi@rsadvnet.it; 4U.O. di Dietetica e Nutrizione Clinica, Policlinico San Martino Hospital IRCCS Hospital-University San Martino, 35122 Genoa, Italy; samir.sukkar@hsanmartino.it; 5International Observatory of Oxidative Stress, Via Paolo Grisignano 21, 84127 Salerno, Italy; eugenioluigi.iorio@gmail.com

**Keywords:** oxidative damage, primary open angle glaucoma, cell cultures, trabecular meshwork NFkB

## Abstract

Glaucoma is a multifactorial disease in which pro-apoptotic signals are directed to retinal ganglion cells. During this disease the conventional outflow pathway becomes malfunctioning. Aqueous humour builds up in the anterior chamber, leading to increased intraocular pressure. Both of these events are related to functional impairment. The knowledge of molecular mechanisms allows us to better understand the usefulness of substances that can support anti-glaucoma therapy. The goal of glaucoma therapy is not simply to lower intraocular pressure; it should also be to facilitate the survival of retinal ganglion cells, as these constitute the real target tissue in this disease, in which the visual pathway is progressively compromised. Indeed, an endothelial dysfunction syndrome affecting the endothelial cells of the trabecular meshwork occurs in both normal-tension glaucoma and high-tension glaucoma. Some substances, such as polyunsaturated fatty acids, can counteract the damage due to the molecular mechanisms—whether ischemic, oxidative, inflammatory or other—that underlie the pathogenesis of glaucoma. In this review, we consider some molecules, such as polyphenols, that can contribute, not only theoretically, to neuroprotection but which are also able to counteract the metabolic pathways that lead to glaucomatous damage. Ginkgo biloba extract, for instance, improves the blood supply to peripheral districts, including the optic nerve and retina and exerts a neuro-protective action by inhibiting apoptosis. Polyunsaturated fatty acids can protect the endothelium and polyphenols exert an anti-inflammatory action through the down-regulation of cytokines such as TNF-α and IL-6. All these substances can aid anti-glaucoma therapy by providing metabolic support for the cells involved in glaucomatous injury. Indeed, it is known that the food we eat is able to change our gene expression.

## 1. Introduction

Glaucoma is a neurodegenerative disease that causes progressive optical damage as a result of the apoptosis of retinal ganglion cells (RGCs) and axon atrophy and degeneration, which extends to the visual areas of the brain cortex and finally leads to the characteristic optical-cup neuropathy and irreversible visual loss (Figure 1). Many factors, including aging, genetic predisposition, exogenous environmental and endogenous factors, play a role in the onset and development of glaucoma. Moreover, several mechanisms are triggered, leading to retinal ganglion cell death due to oxidative stress [1]; inflammation [2], due to mitochondrial dysfunction, endothelial dysfunction [3] and hypoxia [4]; excitotoxicity [5] due to glial dysfunction; and altered axonal transport [6]. In this scenario, glaucoma therapy is aimed solely at reducing intraocular pressure (IOP) and the only drugs or substances commonly used are those which can achieve this. Evidence of an association between diet and glaucoma is not yet clear. However, the intake of nitric oxide, which is present in the leaves of some plants, is reported to be beneficial, while the intake of selenium and iron has a worsening effect [7]. The dietary intake of vitamins A and C has also proved beneficial in glaucoma [8]. Moreover, the role of nutraceuticals in human pathology is evident [9], also at the ocular level [10,11]. This article focuses on some substances that could support anti-glaucoma therapy in counteracting glaucomatous damage. Given the vastness of the subject, we have limited ourselves to presenting only some substances that may be of interest to the reader; we apologize for not being able to include some substances that have known mechanisms and which are already used.

## 2. Glaucoma Pathogenesis

Glaucoma has some target tissues: the Retinal Ganglion Cells (RGCs) of the Optic Nerve Head (ONH); the neuronal chain from the lateral geniculate nucleus to the visual cortex; and the trabecular meshwork (TM) and Schlemm’s canal (SC), which constitute the conventional aqueous outflow pathway (CAOP) (Figure 1). The CAOP is formed by endothelial cells: these act as a barrier [14,15] and are able to change their shape in order to change their gene expression [12]. Therefore, the TM has a real motility that is due to the muscle fibres of the ciliary body, which are able to modify the spaces between the cells and also to change the number of cells exposed to the aqueous humour [12]. Interestingly, the overexpression of NO produces the enzyme endothelial nitric oxide synthase (eNOS), resulting in increased outflow facility and/or decreased IOP [16,17]. This is probably because NO binds the maxi-K channels and leads to relaxation of the TM, most likely by changing the conformation of cytoskeletal proteins [18]. During the course of glaucoma, TM motility is impaired [19].

The TM shows striking morphological decay, its cellularity diminishing in a linear manner with age, with a 47% reduction in the number of cells at 81 years of age [20]. During glaucoma, by contrast, the loss of cells occurs in a gradient-like manner, with the inner tissues being most affected and the outermost tissues least affected [21]. This process *per se* justifies the appearance of glaucoma. Therefore, the genotype of senile trabecular cells is markedly increased [22] and, thus, age is a major risk factor for glaucoma [23]. It should be noted that, with age, the resistance to outflow increases [24] and, in the glaucomatous CAOP, elevated senescence-associated beta-galactosidase (SA- β-Gal) cells are present [25]. The senescence phenotype is associated with endothelial barrier dysfunction [26]. Cells with this particular phenotype may be the result of exposure to different types of stress factors [27], in particular to an oxidative environment [28]. 

The human eye is constantly exposed to sunlight and artificial lighting. Ultraviolet rays are able to alter membranes, nucleic acids and cellular functions. They can also activate pathways that lead to inflammation. In the eye, ultraviolet light does not directly reach the anterior chamber angle. However, the CAOP is more susceptible to oxidative damage than other tissues of the anterior chamber [29].

Oxidative damage, as measured directly on the TM, is much greater in glaucomatous subjects and is directly proportional to IOP and also to visual field defects [30]. Furthermore, visual-field sensitivity appears to be related to a lower systemic antioxidant capacity, as measured by iron reduction activity [31]. Oxidative DNA damage in the TM has been significantly correlated with age and reduced autophagic activity plays a primary role in age-related diseases [32]. In the course of glaucoma, the TM can be compared to a tissue that has aged greatly: there is a significant relationship between oxidative DNA damage and autophagy activation, which is a lysosomal degradation pathway that is essential to the survival and homeostasis of TM cells [33]. Chronic exposure to oxidation leads to lysosomal basification and insufficient proteolytic activation of lysosomal enzymes and consequently to decreased autophagic flux. This might be one of the factors underlying the progressive age-related cell-function failure in the TM, which might contribute to the pathogenesis of primary open-angle glaucoma [34].

In the conventional outflow pathway, the mitochondrial deletion that occurs during glaucoma is much greater than in healthy patients. This alteration occurs only in primary open-angle glaucoma (POAG), in pseudoexfoliative glaucoma [35] and in primary congenital glaucoma [36]. An increase in ROS that exceeds the antioxidant capacity of the tissue results in oxidative stress, contributing to the aging process through the induction and further progression of cellular senescence. The defective mitochondrial function in the TM cells of patients with glaucoma renders these cells abnormally vulnerable to Ca++ stress, with subsequent failure of IOP control [37]. Conversely, the increased expression of Sirtuin 1 (SIRT1) antagonizes the development of oxidative stress-induced premature senescence in human endothelial cells [38]. SIRT1 is a member of the sirtuin family of nicotinamide adenine dinucleotide (NAD+)-dependent histone deacetylases; it helps to regulate the lifespan of several organisms and may provide protection against diseases related to oxidative stress-induced ocular damage [39]. In the case of glaucoma, this is likely to occur through the interaction of SIRT1 with eNOS [40]. Indeed, eNOS activity in HTM cells regulates inflow and outflow pathways [41] and the regulation of eNOS is, in turn, influenced by the activation of Rho GTPase signalling [42] in the AH outflow pathway; this influences actomyosin assembly, cell adhesive interactions and the expression of ECM proteins and cytokines in TM cells in a cascade-like manner [13]. Thus, oxidative stress causes alterations of DNA and RNA and, as a result, will occur in protein and microRNA (MiRNA) [43].

MiRNAs are recognized as important post-transcriptional regulators of gene expression and are known to modulate cellular functions relevant to the normal and pathological physiology of the TM [44]. The chronic damage to the TM that occurs during open-angle glaucoma is reflected in the protein expression of the trabecular cells that flow in the aqueous humour [45]. These proteins accurately reflect the cascade of events that first leads to malfunction of the TM and then to IOP increase. These proteins behave differently in the posterior segment. Indeed, they can become pro-apoptotic signals for the optic nerve head. For instance, in the anterior chamber, AKAP 2 reflects impaired TM motility, while in the posterior segment, an increase in this protein may be an intracellular signal that leads to RGC death by apoptosis [43] (Figure 2). The health of these cells may be compromised by many types of damage; in glaucoma, what probably occurs first is oxidative damage, which triggers the so-called chain of glaucomatous events that lead to the progressive morpho-functional degradation of this pathway. Pro-apoptotic signals are probably the result of the cell dysfunction occurring in the TM and in Schlemm’s canal. Protecting these cells might therefore improve the clinical course of glaucoma. Indeed, the foods that we consume every day are able to alter gene expression in cells, thereby exerting a beneficial or harmful physiological effect. Nutrients therefore play a key role in eye health [46]. For example, from what we have said about the pathogenesis of glaucoma, the well-known substance ginseng has a modulating effect on autophagy, which reduces oxidative stress and improves mitochondrial functions [47]; this substance could therefore be used as an adjuvant in glaucoma therapy.

## 3. Oxidative Stress

Oxidative stress is a physiological adaptive mechanism that most living organisms, from bacteria to humans, exploit in order to respond to environmental challenges, the ultimate aim being to survive. Unlike common stress, which – in humans - is mediated by the hypothalamus, hypophysis and adrenal glands, oxidative stress is under the control of the so-called redox system. The redox system plays a key role in cell homeostasis and survival by modulating cell signalling, defence and detoxification. There is a profound difference between physiological or oxidative eu-stress and pathological or oxidative di-stress. Oxidative di-stress is an emerging health risk factor, which is related to early aging and most common diseases, particularly glaucoma, although it is not always clear whether it is the cause or the effect of each disease. Oxidative stress does not display any clinical picture. 

The antioxidant “family” includes a number of enzymes (e.g., superoxide dismutase, catalase, peroxidase and thioredoxins) and exogenous compounds (vitamins and vitamin-like antioxidant compounds, such as polyphenols, oligoelements, etc.) [48] that exert preventive, radical-scavenging, repair or adaptive functions [49]. These substance are uniformly distributed inside a living organism at both the extracellular and intracellular level [50]. In the extracellular compartment and, particularly, in the blood plasma, all the compounds potentially able to exert a reducing effect that can counteract the “electron avidity” of free radicals constitute the antioxidant plasma barrier. This barrier includes plasma proteins (e.g., albumin), bilirubin, uric acid, cholesterol and all the exogenous, dietary or pharmacological, antioxidants (e.g., ascorbate, tocopherol, polyphenols and so on) [51,52]. In this regard, thiol compounds play a crucial role in ROS modulation [53]. Inside the cell, the antioxidant defence system is well distributed in several compartments. Because the majority of free radicals are generated in lipid layers, which are the sites of the enzymes necessary to catalyse radical-producing reactions, the lipophilic antioxidants (i.e., ubiquinol, vitamin E and beta-carotene) located in biomembranes constitute the first line of defence against ROS; the second line includes water-soluble vitamin C, several members of the vitamin B group and so forth [54].

Interestingly, ROS themselves can stimulate the production of antioxidants, as evidenced by the Nrf-2 system, which provides an excellent example of signal transduction through the involvement of DNA [55]. Moreover, redox signalling molecules derived from the Kreb’s cycle can modulate epigenetic adaptation through histone changes.

The redox system is differently expressed in the various tissues, organs, apparatuses and systems, according to their differentiation. For instance, lens cells show the highest levels of glutathione found in the body, while coenzyme Q10, being abundant in the mitochondria, can be found in many nucleated cells other than red blood cells. This different distribution of redox system components in the body may explain why some organs are more susceptible to oxidative di-stress than others. For instance, it is well know that the primary targets of oxidative di-stress are the cardiovascular apparatus (especially the endothelium), the nervous system (and hence the eye) and the skin. 

The nervous system has a particularly differentiated redox system, with different features in the neurons and in the glia. Neurons are particularly prone to oxidative distress because of: (i) their prevalent aerobic metabolism (high content of mitochondria and hence high probability of ROS generation); (ii) the high level of (oxidizable) unsaturated fatty acids (omega-3 and omega-6) in their cell membranes; (iii) their relatively high availability of free iron (which catalyses hydroperoxide breakdown to highly reactive ROS, according to the Fenton reaction); and (iv) a generally low ability to accumulate vitamin E and coenzyme Q10 in their membranes. This basic knowledge can help us to understand the “metabolic” origin of the ROS that are often detected in neurodegenerative disorders. On the other hand, cell (micro)glia shows a reactive-like phenotype similar to that of leukocytes/monocytes, owing to the abundance of NADPH oxidase, a new potential target of pharmaco-redoxomics. In other words, glia activation may explain the “inflammatory” origin of ROS, which are often associated to neurodegenerative disorders. Therefore, oxidative di-stress is generally recognized to play a pathogenic role in early aging and in several inflammatory and/or degenerative diseases, including atherosclerosis and hypertension (and their consequences, such as stroke and myocardial infarction), Alzheimer’s disease, Parkinson’s disease, cancer [56] and glaucoma [57]. Indeed, so-called silent inflammation or meta-inflammation, is the basis of the so-called pandemics of the third millennium: metabolic syndrome, atherosclerosis, Parkinson’s disease, cancer, infertility, iatrogenic pathologies, ophthalmic lesions and premature aging (oxy-inflammation, inflammaging) [58].

The nervous part of eye, that is, the retina, derives from the brain; indeed, the optic nerve is not a classical cranial nerve; rather, it is like a “protrusion” of the brain into the orbit. The eyes are particularly prone to oxidative di-stress: the anterior part because of its exposure to radiations, the posterior part because it is made up of nervous cells, in which the ROS excess derives either from neurons or from glia.

According to this concept, exogenous and/or endogenous stimuli would trigger the production of ROS, which would activate/inhibit specific biochemical pathways inside the cells, thus allowing them to face environmental changes. For instance, the ROS hydrogen peroxide has proved able to induce reversible oxidative changes of protein thiols, thus modulating key processes involved in cell homeostasis and survival [53,59]. A specific example of a ROS-mediated mechanism of adaptation is seen in the production of oxidants (e.g. hydrogen peroxide and hypochlorous acid) by inflammatory cells in order to protect the tissues against bacterial infections [60]. Another relevant example of physiological modulation by reactive oxidizing species is provided by the nitric oxide pathway in endothelial cells [61].

## 4. Polyunsaturated Fatty Acids

The polyunsaturated fatty acids (PUFA) designated by the abbreviation omega 3 (or *n*-3) are so-called because the double bond closest to the methyl end is separated from it by 3 carbon atoms, which determines the secondary structure that influences their physical properties. Omega-6 and omega-3 fatty acids are essential nutrients and important structural components of cell membranes and are preferentially incorporated into phospholipids [62]. PUFA are able to influence cell membrane properties, such as fluidity, flexibility, permeability and the activity of membrane-related enzymes, transport functions, ion-channel formation and receptors that control metabolite degradation and signals (hormones) between and within cells [63]. Blood levels of omega-3 and omega-6 fatty acids may differ by as much as 30% among individuals; indeed, blood concentrations of fatty acids reflect both dietary intake and biological processes [64]. Docosahexaenoic acid (DHA) is selectively incorporated into retinal cell membranes and postsynaptic neuronal cell membranes [65]. When omega-3 fatty acid intake is low, the retina conserves and recycles DHA, particularly within the outer segments of the rod photoreceptors, where it is found at its highest concentration [66]. Thus, DHA plays an important role in the regeneration of the visual pigment rhodopsin [67]. The phospholipids of neural membranes also contain high proportions of DHA, which is localized in synaptosomes [68]. DHA is able to influence very important processes in the brain, such as synaptic plasticity, neurogenesis, synaptogenesis and neurite growth [62]. PUFA are important for neuronal membrane integrity and function and also contribute to the prevention of brain hypoperfusion resulting from obesity or cerebrovascular diseases [69].

There are several reasons for believing that the use of polyunsaturated fatty acids has a rationale in adjuvant therapy in glaucoma. The first is that they exert a highly protective effect on endothelial cells [70,71]. The aging endothelium shows significant changes in function, as do the cells that compose the conventional outflow pathway. This could be partly due to high levels of oxidative stress and ischemia. Adenosine monophosphate–activated protein kinase (AMPK) regulates cellular metabolism, proliferation and aging processes [72]. AMPK is the main energy sensor of all living cells and is involved in neuronal activity [73]. Moreover, AMPK is protective against myocardial ischemia [74] and in the entire heart [75]. In addition, this molecule reduces oxidative stress in brain injury through the modulation of the signalling type 2 cannabinoid receptor [76]. Indeed, the activation of AMPK inhibits the production of ROS induced by mitochondrial dysfunction, ER stress and NADPH oxidase. Finally, AMPK inhibits the production of pro-inflammatory factors, preventing endothelial dysfunction by increasing the bioavailability of nitric oxide [77]. In the eye, AMPK has proved to be a critical regulator of ECM homeostasis and cytoskeletal arrangement in the TM [78], both of which are of relevance to glaucoma, the former because the TM is organized into a network of beams, with ECM/matrix occupying the spaces between the beams [79] and the latter because the motility of the TM is fundamental to its functioning [12]. One of the molecular mechanisms by which the acids EPA and DHA prevent ischemic injury is the activation of extracellular signal-regulated AMPK and endothelial nitric oxide synthase (eNOS) [80]. Omega-3 fatty acids also reduce blood viscosity [81], probably because they improve the deformability of the red blood cells [82].

Several inflammatory molecules are up-regulated during glaucoma (Figure 3). These include vascular endothelial growth factor (VEGF), interleukins and tumour necrosis factor alpha (TNF-α) [83]. Hypoxia and NO are activators of VEGF, which is able to influence the optic axon, though the mechanism is not yet clear [84]. In the anterior segment, increased levels of VEGF have been observed, particularly in the TM, where its presence could be linked to the presence of other factors, such as interleukins, all of which are directly/indirectly involved in the TM tissue remodelling process [85]. Indeed, during glaucoma, all these cytokines can induce ECM remodelling and alter cytoskeletal functions in the TM [86]. Specifically, IL6, which increases as a result of oxidative stress, is able to improve perfusion in porcine eyes and is also implicated in the induction of senescence [87] and in the modulation of barrier functions of the endothelium of the TM [88], while IL8 is able to modulate the permeability of the endothelial cells of Schlemm’s canal [89].

TNF-α is a pro-inflammatory cytokine that is also able to perform neuroprotective functions. Indeed, this cytokine acts through two distinct receptors: TNF-R1 and TNF-R2 and its activity changes according to the activation of one or the other of these 2 receptors. The first receptor, if activated, leads to the recruitment of immune cells, causing inflammation (Figure 3) and can also activate enzymes that induce oxidative stress [93]. The second one, by contrast, plays an active role in neuroprotection, as it supports tissue homeostasis and promotes tissue regeneration [94]. This effect is dependent on the activation of the NFkB pathway by the stimulation of TNF-R2, which translocates into the nucleus of cortical neurons, thus presumably exerting a protective effect against excitotoxicity [95]. However, it should be emphasized that in vitro experiments have demonstrated that TNF-α can induce RGCs apoptosis via caspase-3 and -8 activation or through oxidative stress induced by mitochondrial dysfunction [93]. Furthermore, increased ROS formation can directly determine neuronal cell death. The ROS activating the NFkB pathway in the glial cells activates inflammation, which in turn activates NADPH oxidase, which in turn leads to the production of ROS, thus triggering a vicious circle (Figure 3).

In this context, Omega-3 polyunsaturated fatty acids, which have well-documented anti-inflammatory properties, may have therapeutic potential in chronic inflammatory diseases, such as glaucoma. Indeed, Wang et al. [96] found that increasing the daily dietary intake of PUFA, including ω-3 fatty acids, was associated with a significant decrease in the probability of glaucoma. Finally, oral omega-3 supplementation for 3 months has been seen to significantly reduce IOP in normotensive adults [97] and in pseudoexfoliative glaucoma [98].

Another substance that has anti-inflammatory properties is quercetin; this flavonoid can be isolated from many plant species, including Ginkgo Biloba, which we will discuss below. Baicalin is also a known bioflavonoid, the neuroprotective efficacy of which seems to be strictly related to its anti-inflammatory and antioxidant functions, as well as its ability to protect the mitochondria and to inhibit glutamate-induced neurotoxicity [99]. Flavonoids are widespread in nature, being found in a vast range of plants, including citrus fruits, grapes, tomatoes, berries and green tea; more than 5000 compounds that exert beneficial effects on health are known. An emblematic example is ginkgo biloba.

## 5. Ginkgo Biloba Extract

Ginkgo is an ancient species of tree that has features in common with plants that lived 270 million years ago. Extracts from the leaf of the maidenhair tree, *Ginkgo biloba*, have been used to treat pulmonary disorders in Chinese traditional medicine since 3000 BCE [100] and are used worldwide as a herbal medicine [101]. Ginkgo biloba extract (GBE) is the most widely used drug in Germany. In the USA, it is a freely available nutritional supplement. In modern medical science, GBE is indicated for the treatment of cognitive impairment and dementia [102] but has also proved effective in diseases associated with aging, such as vascular disease, tinnitus, asthma and erectile dysfunction. As glaucoma and Alzheimer’s dementia display many biological and mechanistic similarities [103], some authors have investigated the possibility of using GBE to treat glaucomatous optic neuropathy. Indeed, many of the properties of GBE can be useful in the treatment of non-IOP-dependent risk factors in glaucoma [104].

GBE is mainly composed of flavonoids such as quercetin and rutin and terpenoids such as bilobalide and ginkgolides [105,106]. Several studies on experimental systems and tissues have demonstrated the efficacy of GBE in reducing free-radical damage and lipid peroxidation [107,108,109,110,111,112] and some have demonstrated the benefits of EGb761 in Alzheimer’s, vascular dementia and cognitive impairment [113,114,115]. Various preclinical studies have also demonstrated the neuroprotective effects of EGb761 [116,117,118]. Currently, EGb761 seems to prevent neurodegenerative dementias associated with aging, Alzheimer’s Disease (AD), peripheral vascular diseases and neurosensory problems (e.g., tinnitus) and its use is common in these disorders [119,120]. 

One of the possible properties of GBE is that it may increase blood flow, perhaps by reducing the viscosity of the blood. Indeed, one study found that it increased perfusion in skin and nail bed capillaries, prompting the authors to speculate that it might lower blood viscosity without changing laboratory coagulation parameters [121]. GBE should not be taken by patients on anticoagulant therapy [122], as it may interact with anticoagulant and antiplatelet agents [123]. Indeed, GBE seems to be associated with spontaneous bleeding (e.g., sub-arachnoid haemorrhage [124,125], subdural hematomas [126] and hyphema [127] and strongly inhibits platelet-activating factor (PAF) [128,129], which promotes platelet aggregation, neutrophil degranulation and oxygen radical production [130]. Moreover, oxidative stress can raise the concentration of PAF [131], elevated levels of which have been found in experimental ischemic cerebral injury [132]; it may also increase glutamate excitotoxicity in brain injury [133]. By contrast, GBE has been seen to prevent glutamate neurotoxicity in a murine model [134]. 

The retina is endowed with PAF receptors and PAF is reported to increase glutamate release in the retina, too [135]. However, GBE treatment has been seen to exert positive effects on PAF-induced ERG disturbances in a rat model [136]. Thus, GBE may be able to inhibit glutamate release, thereby providing neuroprotection in glaucomatous optic neuropathy [137]. It has also been shown experimentally that GBE protects against lipid peroxidation in erythrocytes [138] and that the inhibition of lipid peroxidation by GBE protects the vascular endothelium [139] and rat spinal cord [140]. GBE can also rid the tissues of free radicals [141,142]. Some studies have investigated the direct ocular effects of GBE. In a randomized, prospective, double-blind, cross-over trial in normal eyes, oral GBE improved ocular blood flow, without affecting blood pressure, heart rate or IOP [143]. Other ocular effects, such as reduced retinal degeneration after intravitreous injection of proteolytic enzymes, have been seen in rabbits [144]. Indeed, again in the rabbit eye, the intravitreous injection of GBE reduced the induced vitreoretinal proliferation, perhaps through radical scavenging [145]. In another study, GBE seemed to improve colour vision in patients with diabetic retinopathy [146]. Systemically, GBE has proved useful in AD [147], in terms of functional and cognitive assessment. As visual field variability is affected by alertness and fatigue [148,149], it is possible to speculate that improving the neurological condition could have a positive but reversible, effect on the visual field [150].

The role of mitochondrial dysfunction in the pathogenesis of glaucoma has now been clarified by a large number of studies [151]. Because only anti-oxidants are able to penetrate into the mitochondria, they may be useful as neuroprotective agents. Some of the molecules contained in GBE, such as polyphenolic flavonoids, may help to prevent oxidative stress in the mitochondria and preserve RGCs [30,103,152]. Accordingly, in normal-tension glaucoma (NTG) patients, oral GBE therapy has been seen to improve visual field defects and visual field indexes [153]. In another study, however, GBE was not able to improve contrast sensitivity or visual field damage in Chinese patients with NTG [154]. Both of these studies were limited by their short follow-up period and small sample size. In any case, GBE displays a relatively safe profile [155] and, despite the inconclusive nature of clinical studies on its neuroprotective effect, some glaucoma specialists have been prescribing GBE for their patients as an adjuvant therapy for many years [156]. 

However, the reported increased risk of bleeding during surgery can increase the risk of adverse events [157]. Nevertheless, a daily dose of 120 mg of oral GBE is reported to be highly efficacious and safe [155]. Moreover, the administration of GBE is associated with a reduction in IOP in most glaucoma patients and with a lower economic burden. However, GBE can be considered a valuable tool only in patients affected by NTG; in those with high-tension glaucoma (HTG) it can be used only if the disease is in remission and IOP has been adequately reduced [156].

Medical, laser and surgical treatments for glaucoma throughout the twentieth century focused on methods of lowering IOP. However, despite successful lowering of IOP, many patients continue to suffer progressive damage, which implies the presence of non-IOP-dependent risk factors. In addition, some glaucoma patients do not have elevated IOP: that is, they have NTG. Nevertheless, the damage to the visual field is similar in both types of glaucoma, even though the pathogenic factors involved are considerably different and their topography is also different. Indeed, localized nerve fibre layer defects in NTG are reported to be closer to the fovea and of greater width than those in POAG [158]. Vascular or perfusion abnormalities in NTG include the increased frequency of migraine headaches, Raynaud’s phenomenon and sleep apnoea [159]. In NTG, RGC death is not determined by IOP increase; other factors are of pathogenetic relevance. Indeed, blood vessel injury in glaucoma may contribute to altered local blood flow modulation and the vascular theory of glaucoma pathogenesis could explain cases of NTG [160]. Retinal ischemic injury results in a self-sustained cascade involving neuronal depolarization, calcium influx and oxidative stress; this cascade is triggered by an energy deficit and an increase in glutamatergic stimulation [161], leading to the loss of RGCs [162,163,164]. Values of optic nerve head blood flow decrease with increasing age [165]. Normal-tension glaucoma increases in prevalence with age [166] and is associated with ocular hemodynamic abnormalities [167]. Elevated IOP is accompanied by a reduction in ocular blood flow parameters [168,169,170,171], while reducing IOP improves these parameters [172]. Optic nerve sheath decompression improves visual function in patients with papilledema [173] and anterior ischemic optic neuropathy [174]. These changes have been attributed to improved blood flow velocities and decreased resistance in the central retinal artery and posterior ciliary arteries.

Numerous investigators have examined the effects of IOP-lowering drugs on ocular blood flow in humans [175,176,177,178,179,180,181]. Both Chinese herbal preparations and acupuncture [182] have been reported to improve ocular blood flow. More recently, the application of neuroprotection to the treatment of non-IOP-dependent glaucomatous damage has received increasing attention. In addition to the greater awareness of the extent of normal-tension glaucoma in the population, it has been seen many patients with high-tension glaucoma continue to suffer progressive visual field loss after normalization of IOP [183,184]. It has been postulated that excitatory amino acids, particularly glutamate, play a role in the progression of glaucomatous damage at normal levels of IOP [185]. Larger ganglion cells are particularly susceptible to glutamate toxicity [186]. Elevated glutamate levels have been found in the vitreous of glaucoma patients, with higher levels in patients with worse disease [187]. It is still uncertain, however, whether high vitreous levels of glutamate are a cause or a result of damage or both but such concentrations are sufficient to cause significant damage to the RGCs. 

The progressive loss of nerve cells constitutes a general pattern of damage in neurologic diseases, so that, regardless of the primary cause of neuronal cell death, damage spreads beyond directly injured neurons to adjacent neurons that escape the primary lesion [188,189]. Whether the primary lesion is caused by hypoxia–ischemia, stroke, seizure disorders, trauma or degenerative disease, changes in the extracellular environment include alterations in ion concentrations, increased free radical concentrations, neurotransmitter release, depletion of growth factors and immune system involvement [188,190]. These changes lead to apoptosis [191,192,193,194]. Although neuroprotective strategies and pharmaceutical agents have been initiated in the treatment of numerous disorders of the central and peripheral nervous systems, including trauma, epilepsy, stroke, Huntington’s disease, amyotrophic lateral sclerosis, AIDS and dementia, none have yet been applied to the treatment of glaucoma. GBE has numerous properties which, theoretically, should be beneficial in treating non-IOP-dependent mechanisms in glaucoma. Its multiple beneficial actions, including increased ocular blood flow, antioxidant activity, platelet-activating factor inhibition, nitric oxide inhibition and neuroprotective activity, combine to suggest that GBE could prove to be of major therapeutic value in the treatment of glaucoma. 

## 6. Polyphenols

Polyphenols are plant-derived organic substances, the chemical structure of which requires the presence of at least one phenol, which may be substituted in various ways. Polyphenols can be divided into different subclasses, according to the number of phenolic rings present in their structure. They comprise 4 families: simple phenolic acids (e.g., caffeic, ellagic and vanillic), stilbenes (e.g., resveratrol), coumarins and flavonoids. Polyphenols are very common in nature and their distribution is almost ubiquitous. All these substances display poor intestinal absorption, which conditions their bioavailability. Moreover, they have several biological properties, including the ability to chelate toxic and transition metals, to modulate the distribution of bacterial populations comprising the intestinal microbiota, to induce the synthesis of endogenous antioxidants through the transcription factor Nrf2 and to inhibit the NFkB factor [57]. Interestingly, they are also associated with the mechanism of Nuclear factor erythroid-2-related factor 2 (Nrf2), a transcription factor that is up-regulated in times of oxidative stress. Indeed, this factor protects the cell from oxidative stress. In the absence of oxidative stress, Nrf2 remains in the cytosol but in the presence of ROS the Keap1 protein allows Nrf2 to translocate into the nucleus, activating the transcription of antioxidant enzymes by binding to the antioxidant response element in the promoter regions of its target genes [195]. Nrf2 is involved in protection against diseases due to oxidative stress [196] and particularly in neurodegeneration [197]. Indeed, intracellular ROS generated in the mitochondria play an important role in the activation of Nrf2 and up-regulation of antioxidant and detoxification systems [198]. Furthermore, Nrf2 has proved to be useful in protecting RGCs against oxidative damage in vivo [199] and constitutes an important cytoprotective mechanism in the retina in response to ischemia-reperfusion injury [200], thereby exerting a real neuroprotective function [201]. It is interesting that, in glaucomatous cells of the TM, Nrf2 is down-regulated in comparison with healthy TM cells and could play a role in the apoptosis of trabecular cells by regulating apoptosis-related proteins [202]; this effect is probably due to microRNA-93, which is able to inhibit the expression of Nrf2 in TM cells in glaucomatous subjects [203].

Another important effect of polyphenols that has implications for glaucoma is their impact on mitochondria. The important role of mitochondrial damage in POAG is supported by the finding that myocilin, which is the product of a gene that has been linked with open-angle glaucoma [204], alters mitochondrial functions in human trabecular cells [205]. More specifically, during glaucoma, TM cells with up-regulated myocilin can be prompted to trigger the depolarization and death of mitochondria [206]. Glaucoma is now known to be associated with decreased mitochondrial respiration, increased mitochondrial ROS production, dysregulation of mitochondrial biogenesis, decreased mitophagic signalling and increased apoptosis [207,208]. One extremely important factor is the peroxisome proliferator-activated receptor gamma coactivator 1 family (PGC-1), which in many tissues [209] organizes mitochondrial biogenesis and the expression of mitochondrial transcription factor A (TFAM), a key factor that controls mtDNA replication and transcription [210].

Many polyphenols have the ability to activate Sirtuin 1 (SIRT1), which in turn activates PGC-1 [211]. Substances such as resveratrol or quercetin are able to exert these functions. Specifically, resveratrol is able to increase mitochondrial mass and mitochondrial DNA content in endothelial cells by up-regulating endothelial nitric oxide (NO) synthase (eNOS) in a SIRT1-dependent manner. Indeed, in one study, SIRT1 was induced and endothelial nitric oxide (NO) synthase (eNOS) was up-regulated [212]. Curcumin is also able to increase PGC-1 levels by improving mitochondrial membrane potential (MMP) and ATP levels in the brain of senescent mice [213]. Curcumin is an antioxidant, in that it induces the expression of cytoprotective proteins or antioxidant enzymes such as superoxide dismutase, catalase, glutathione reductase, glutathione peroxidase and glutathione-S-transferase [214]. Moreover, polyphenols such as resveratrol and epigallocatechin-3-gallate activate detoxifying and antioxidant enzymes by means of the common transcription factor Nrf2 [215].

Polyphenols derived from red wine, tea and dark chocolate regulate vascular reactivity by targeting endothelial nitric oxide synthase (eNOS) and inducing nuclear accumulation of Nrf2 [216]; this enhances the bioavailability of NO [217], which, by reacting with superoxide anions to form peroxynitrite, up-regulates the nuclear accumulation of Nrf2, thus protecting RGCs cells from glycation-induced neurotoxicity. Finally, polyphenols reduce inflammation through a number of mechanisms, such as the reduction of the expression of cytokines such as IL-2 IL-6 and TNF-alpha. This is associated with the activation of the transcription factor NF-κ B, which regulates the release of these mediators of inflammation [218] (Figure 3). Curcumin reduces the systemic and tissue inflammatory response by preventing the release of TNF-α and C-reactive protein [219]. 

## 7. Conclusions

Epidemiological studies have shown that various fruits and vegetables can act as chemo-protective agents, as the different substances contained in them can interact with the pathogenetic mechanisms of many diseases. Indeed, they are able to regulate several cellular molecular pathways involved in the regulation of inflammation, redox potentials, metabolic disorders and apoptosis. Many natural substances and the metabolic pathways and interactions in which they are involved deserve to be investigated. Much more could be said about the substances that we have briefly described. However, the purpose of this review is limited to arousing interest in the possible use of some of these in the adjuvant therapy of glaucoma. 

Glaucoma is still a challenge for researchers, not only on account of its complexity but also because of all the substances and metabolic pathways involved. Moreover, the target cells in the eye which are involved in the pathogenesis of glaucomatous disease are very varied; for this reason, simply reducing IOP is not sufficient to guarantee a good prognosis in this disease. The use of adjuvants to counter the basic mechanisms underlying the pathogenesis of glaucoma therefore appears to be a goal to pursue.

## Figures and Tables

**Figure 1 nutrients-11-00239-f001:**
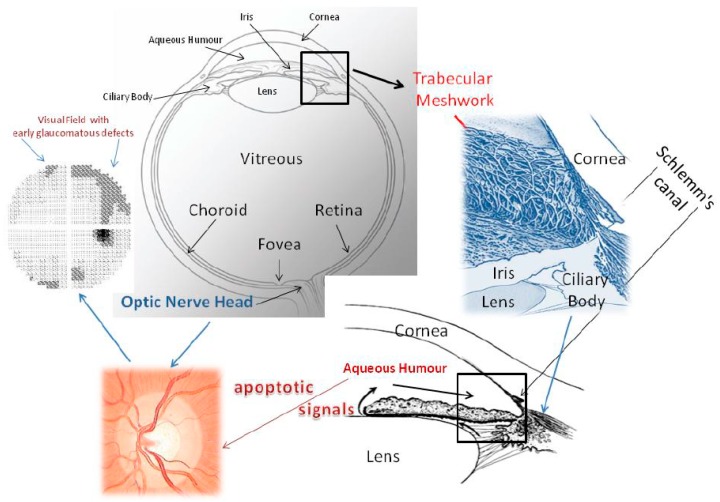
This picture shows the whole eye and the iridocorneal angle, seen both two-dimensionally and three-dimensionally, to explain the aqueous humour outflow pathway. Malfunctioning of the trabecular meshwork causes IOP to increase. The conventional outflow pathway is probably the tissue which is responsible for the proapoptotic signals that determine alterations of the optic nerve head and of the visual field. The trabecular meshwork consists of endothelial cells immersed in their fundamental substance. Aqueous humour flows through the intercellular spaces of the TM and crosses the inner wall of Schlemm’s canal. From a functional point of view, the conventional aqueous outflow is endowed with two barriers. The first is formed by the trabecular meshwork cells and the second by the endothelial cells that line the lumen of Schlemm’s canal. Located between the iris and cornea, it has a particular architecture, which considerably increases the filtration surface; its endothelial cells are constantly in contact with free radicals. Oxidative stress is chiefly responsible for molecular damage to its endothelial cells and triggers all those events that lead to glaucoma. Oxidative attack induces a loss of trabecular meshwork cells, impairing TM functionality. It is not known whether oxidative damage is due to reduced antioxidant defences or to primary damage to mitochondria. In addition, free radicals are implicated in the mechanism of senescence. During the course of glaucoma, the TM displays cell loss, subclinical inflammation, increased accumulation of extracellular matrix, endothelial dysregulation and dysfunction, changes in the cytoskeleton, altered motility and outflow impairment. The aqueous humour proteome profile also undergoes dramatic changes, reflecting cellular and molecular damage to the TM. We still do not know the mechanism that links trabecular damage to the apoptosis of ganglion cells. From a clinical standpoint, the death of ganglion cells causes alterations in the visual field. Furthermore, in glaucoma, the aqueous humour proteins, which are an expression of TM failure, among other things, might constitute biological signals for the posterior segment, where the cascade of events leading to the process of degeneration involves ganglion cells [12,13].

**Figure 2 nutrients-11-00239-f002:**
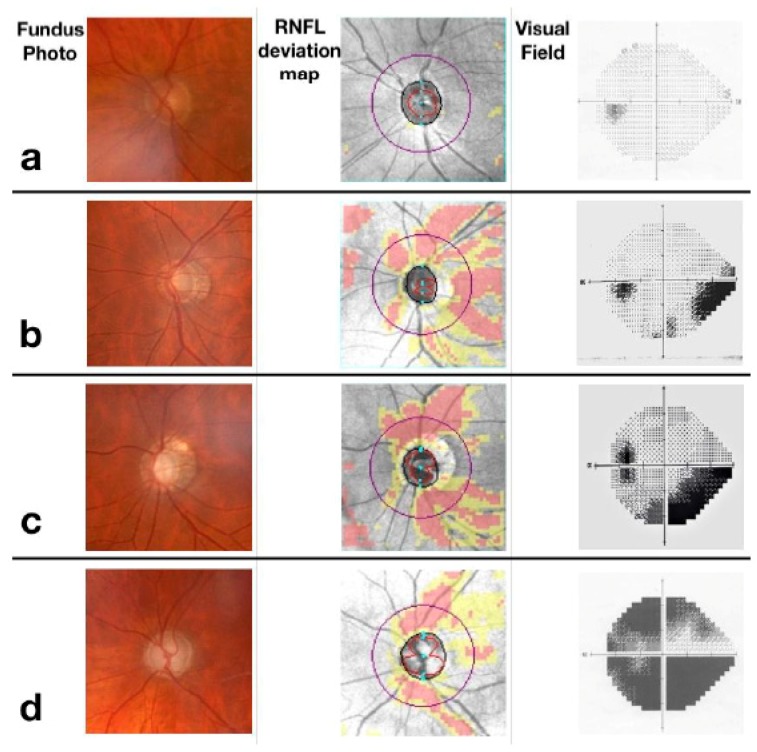
Glaucoma stages. Row (**a**) representation of a normal optic nerve, which presents nerve fibre layers and unaltered visual field. Row (**b**) early-stage glaucoma with initial optic disc excavation at the fundus and loss of nerve fibres more pronounced in the superior sector that determines a nasal step in the visual field. Row (**c**) moderate-stage glaucoma with partial optic disc excavation at the fundus and loss of nerve fibres in the inferior sector, determining an arciform defect in the visual field. Row (**d**) severe glaucoma, characterized by an excavated optic disc, loss of nerve fibres in both superior and inferior sectors and absolute defects in all sectors of the visual field.

**Figure 3 nutrients-11-00239-f003:**
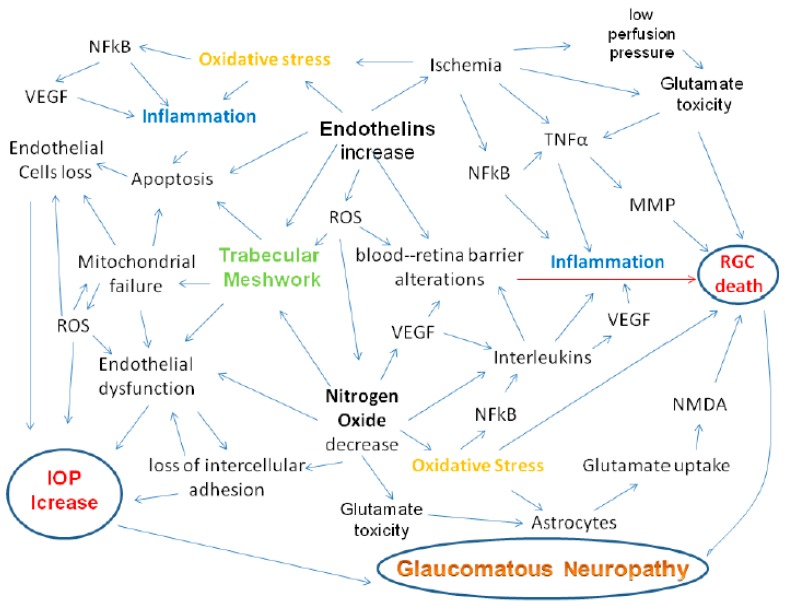
Inflammation due to oxidative stress is the basis both of alterations to the conventional outflow pathway, leading to IOP increase [13] and of the alterations that induce apoptosis of RGCs. Indeed, in glaucoma, the expression of inflammatory genes occurs. ROS also activate transcription factor NF-kB, which induces the expression of various agents, including pro-inflammatory cytokines (IL-1/6, TNF-α) [90]. Pro-inflammatory cytokines, such as TNF-α or interleukins, are up-regulated, inducing intracellular and extracellular ROS production in human RPE cells [91]. Between oxidative stress and inflammation, NF-kB plays a strategic role, entering into the nucleus to induce transcription of a myriad of genes that mediate diverse cellular processes, such as immunity, inflammation, proliferation, apoptosis and cellular senescence [92].

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
