# Peer review of "Substances of Interest That Support Glaucoma Therapy"

_nutrients, 2019, doi:10.3390/nu11020239_

Round 1
Reviewer 1 Report
The paragraph "Glaucoma pathogenesis" is far too long (77-346). Moreover, it does not comply with the abstract. There should be more focus on the nutrients (also when looking at the scope of this journal). This large paragraph should be removed or shortened to<1 page.
Poor quality of pictures. Also, Figure 1 is not clear, especially not for someone without an ophthalmic background.
Table 1 presents a nice overview, but many of the substances are not covered in the review. For example, Ginseng, Baicalin, Quercentina, Citicoline are not mentioned in the text. It makes no sense to mention these nutrients in the Table if they are not covered in the manuscript.
Line 78. Glaucoma had three target tissues -> According to the anatomy it is incorrect to see the ONH and optic nerve as two different structures, therefore, this should be changed to two target tissues instead of three.
Figure 2.: There is a type in the first sentence. Furthermore, the ONH shown in the second column are not the same as the first column. Please do not mix optic disc photo's and RNFL-scans of different eyes.
Line 278 till 284 should be removed. It does not add anything to the manuscript.
Figure 3.: Is a nice overview of pathways, but could be better summarized. The number of arrows are like a maze... Furthermore, please correct the typos...
In conclusion, the topic is very interesting, but the manuscript is poorly organized and should focus more on nutrients rather then pathophysiologic mechanism of which some are not relevant for the mentioned nutrients in the manuscript. Major revisions are required.
Author Response
Dear Editor,
We hereby submit our revised manuscript titled “Substances of interest that support glaucoma therapy” to Nutrients for review.
We have given careful consideration to all comments of the reviewers and have revised the manuscript extensively to address those concerns.
We thank the Editor for the overall interest in our manuscript and topic area. We will address each point individually:
Reviewer 1
The paragraph "Glaucoma pathogenesis" is far too long (77-346)…….
The chapter on the pathogenesis of glaucoma has been very shortened, also because it cannot be completely eliminated because it is not possible to explain the molecular mechanisms of its pathogenesis and to understand the use of the substances mentioned in the article. Furthermore, new elements have been introduced to complete the description of the substances mentioned in the article.
Poor quality of pictures. Also, Figure 1 is not clear, especially not for someone without an ophthalmic background.
In the article there are 3 figures, the first is necessary for those who are not an ophthalmologist to understand, not only how an eye is made but also because its pathophysiology is not simple. This is the reason for such a long legend. To explain better the figure 1 we have added another sentence. The second figure consists of images of examinations performed in clinical practice. Finally, the third one is the schematic representation of the inflammation that occurs in the eye during glaucoma, both in the anterior segment and posterior segment (described in the text of the article). Maybe it's complicated, but that's what happens from a molecular point of view.
Table 1 presents a nice overview, but many of teh substances are not covered in teh review. For example, Ginseng, Baicalin, Quercentina, Citicoline are not mentioned in teh text. It makes no sense to mention these nutrients in teh Table if they are not covered in teh manuscript.
We have added in the text a brief description of all the substances mentioned in table 1.
Line 78. Glaucoma had three target tissues -> According to the anatomy it is incorrect to see the ONH and optic nerve as two different structures, theirfore, dis should be changed to two target tissues instead of three.
We modified the sentence for a better comprehension: the target tissues are: 1° the retinal ganglion cells (RGCs) of the optic nerve head, 2° the whole neuronal chain that reaches the calcarine fissure and 3° the conventional pathway of outflow.
Figure 2. their is a type in teh first sentence. Furthermore, teh ONH shown in teh second column are not teh same as teh first column. Please do not mix optic disc photo's and RNFL-scans of different eyes.
In this picture eyes come from the 3 different patients A, B and C. Tests correspond and belong to the aforementioned patients. In the first column there is the optic disc picture, in the second the RNLF examination and in the third the visual field. Probably the different cam for the imaging of the optic nerve is responsible for the different appearances. If you do not like some patients tell us which one in particular and we will see to replace it with someone else.
Line 278 till 284 should be removed. It does not add anything to the manuscript.
These sentences have been removed from the text.
Figure 3.: Is a nice overview of pathways, but could be better summarized. Teh number of arrows are like a maze... Furthermore, please correct teh typos...
The figure is complex and so organized: the left part concerns the anterior segment and the right side the posterior segment. The events occur from the top to the bottom and from left to right. We know the topic is complex, but we do not know how to make this picture less complex. We are open for any suggestion from the reviewer.
We sincerely thank the Reviewers and the Editors for their time and suggestions. We hope that
with the revisions, the manuscript will now be acceptable for publication.
Sincerely yours,
Sergio C Saccà

Reviewer 2 Report
Authors in this manuscript have reviewed and discussed the molecular mechanisms involved in glaucoma pathogenesis along with the detailed description of the substances that are know to play a role in neuro- protection. In this review, authors are emphasizing on the use of these substance that could support the therapy in counteracting the glaucomatous damage.
Minor Points:
1. Abstract: Use of the term antihypertensive therapy in glaucoma is not appropriate while discussing the management of glaucoma pathogenesis and effective therapy.
2. Abstract is missing coherence and order, e.g. Glaucoma and it’s pathogenesis should be described first and then why understanding the usefulness of substances that can support glaucoma therapy can be mentioned.
3. Retinal ganglion cells are the main cell that degenerate in all kind of glaucoma (independent of IOP) and ultimately leads to blindness. Retinal ganglion cells and the other involved cells (TM and endothelial cells) should be mentioned in continuation of disease pathogenesis.
4. Page 2, line 74: Please change “14 and b” in figure 1 legend.
5. Page 4, line 138: While discussing impaired mitochondrial function within the trabecular meshwork Authors should also include the study done by Tanwar et al., 2010 (Mol Vis. 2010 Mar 24;16:518-33) where authors have reported pathogenic mtDNA mutations in primary congenital glaucoma.
6. Authors can simplify the figure 3 in a way that all the alterations (Oxidative stress, inflammation, IOP etc.) should come in one line and then surrounded by multiple factors like interleukins, gene expression etc. connecting with each other.
Author Response
Dear Editor,
We hereby submit our revised manuscript titled “Substances of interest that support glaucoma therapy” to Nutrients for review.
We have given careful consideration to all comments of the reviewers and have revised the manuscript extensively to address those concerns.
We thank the Editor for the overall interest in our manuscript and topic area. We will address each point individually:
Reviewer 2
Abstract: Use of teh term antihypertensive therapy in glaucoma is not appropriate while discussing teh management of glaucoma pathogenesis and effective therapy.
We have modified the term “antihypertensive” with a more precise term “anti-glaucoma”.
Abstract is missing coherence and order, e.g. Glaucoma and it’s pathogenesis should be described first and then why understanding teh usefulness of substances that can support glaucoma therapy can be mentioned.
We have modified the abstract according to the reviewers suggestions.
Retinal ganglion cells are the main cell dat degenerate in all kind of glaucoma (independent of IOP) and ultimately leads to blindness. Retinal ganglion cells and the other involved cells (TM and endothelial cells) should be mentioned in continuation of disease pathogenesis.
We believe that in case of high-pressure glaucoma the endothelium of the trabecular meshwork is the primum movens of glaucoma and that the ganglion cells death is secondary to pro-apoptotic signals for the degeneration of the conventional outflow pathway. If the referee would like to read some articles on the pathogenesis of glaucoma that we otherwise published in high impact journals of molecular biology, we suggest to read: Saccà et al. From DNA damage to functional changes of the trabecular meshwork in aging and glaucoma. Ageing Res Rev. 2016; 29: 26-41. If the referee wants to speak about specific questions we could answer him. Low-pressure glaucomas, as rightly pointed out by the referee, do not include the trabecular meshwork failure but the role of its endothelium is important. Low-pressure glaucomas, however, at least for the Caucasian race, represent a minority of cases.
Page 2, line 74: Please change “14 and b” in figure 1 legend.
We have modified the legend according to the reviewers suggestions.
Page 4, line 138: While discussing impaired mitochondrial function wifin teh trabecular meshwork Authors should also include teh study done by Tanwar et al., 2010 (Mol Vis. 2010 Mar 24;16:518-33) where authors has reported pathogenic mtDNA mutations in primary congenital glaucoma.
We have added the reference the referee suggested.
Authors can simplify teh figure 3 in a way dat all teh alterations (Oxidative stress, inflammation, IOP etc.) should come in one line and tan surrounded by multiple factors like interleukins, gene expression etc. connecting wif each other.
Trying to represent schematically the relationships between inflammation and glaucoma is not very simple. As already answered to the first referee the figure is complex (See answer to the first referee) and in our opinion simplifying it could cause the reader to underestimate this aspect in the pathogenesis of glaucoma. Moreover, all the molecular pathways shown in the figure are described in the text.
We sincerely thank the Reviewers and the Editors for their time and suggestions. We hope that
with the revisions, the manuscript will now be acceptable for publication.
Sincerely yours,
Sergio C Saccà

Round 2
Reviewer 1 Report
Table 1 does not add anything to the manuscript. Line 89. Glaucoma has three target tissues -> you are now mentioning more than three... Figure 2.: There is still a typo in the first sentence. Furthermore, the ONH shown in the second column are not the same as the first column. Please do not mix optic disc photo's and RNFL-scans of different eyes. -> this has not been corrected! In conclusion, the topic is very interesting, but the manuscript is poorly organized and should focus either on nutrients or the pathophysiologic mechanism. Or, the paper should be split in two separate papers. Furthermore, there are still many typos that should be corrected/errors.
Author Response
Dear referee,
Enclosed please find the revised version of the paper titled “Substances of interest that support glaucoma therapy” sent to Nutrients for review.
The reviewers' comments have been addressed on a point-by-point basis.
Table 1 does not add anything to the manuscript.
We have deleted Table 1
Line 89. Glaucoma has three target tissues -> you are now mentioning more than three...
We have replaced the word “three” with “some”
Figure 2.: their is still a typo in teh first sentence. Furthermore, teh ONH shown in teh second column are not teh same as teh first column. Please do not mix optic disc photo's and RNFL-scans of different eyes. -> this has not been corrected!
The typo has been corrected. With regard to the ONH images shown in Figure 2, as we said in our last reply, the eyes included in the figure belonged to the same patients. The reviewer may have been misled by the fact that different cameras were used. In any case, we have modified the figure, which now shows images from four different patients.
In conclusion, teh topic is very interesting…
Many thanks !
There are still many typos that should be corrected/errors.
The manuscript has been carefully revised by a native speaker of English with almost 30 years of experience in the translation and revision of scientific texts in general and medical papers in particular. If you have spotted any examples of incorrect or obscure language, we would be most grateful if you could point them out to us.
We look forward to receiving further suggestions from the reviewers.
We have amended the text, introduced new subsections and tried to avoid repetitions and to make the figures more readable.
We thank the reviewers for their suggestions and the opportunity to further substantiate our results. We hope that these changes may be useful to the publication of this article.
Sincerely,
Dr. Sergio Claudio Saccà
